# What Is Wrong with Hyaluronic Acid Chemistry? A ^15^N/^13^C Solid-State NMR Re-Evaluation of Its Dopamine Conjugates

**DOI:** 10.3390/polym15132825

**Published:** 2023-06-26

**Authors:** Ioana-Georgeta Grosu, Augustin Moț, Xenia Filip, Claudiu Filip

**Affiliations:** 1National Institute for R&D of Isotopic and Molecular Technologies, 67-103 Donat Street, 400293 Cluj-Napoca, Romania; ioana.grosu@itim-cj.ro (I.-G.G.); augustin.mot@itim-cj.ro (A.M.); xenia.filip@itim-cj.ro (X.F.); 2Department of Chemistry, Babeș-Bolyai University, 11 Arany Janos Street, 400028 Cluj-Napoca, Romania

**Keywords:** hyaluronic acid, dopamine, oxidized hyaluronic acid, solid-state NMR

## Abstract

In this work, a systematic ^15^N/^13^C solid-state NMR investigation is performed on three dopamine (DA) conjugates of hyaluronic acid, considered in both its native (HA) and NaIO_4_-oxidized (HA_Ox_) forms. Two of them, here named HA_EDC_-DA and HA_Ox_-DA, have been previously introduced as covalent conjugates involving DA amine nitrogen: the former by EDC-mediated amide bond formation, and the latter by reaction of the Schiff base with the aldehyde moieties presumed to exist in HA_Ox_. The third conjugate, HA-DA, is reported here for the first time; it is obtained by simply mixing hyaluronan with DA∙HCl at pH 5. The ^15^N ss-NMR spectra were found to be consistent in all the systems, and the DA molecules were found to be in their charged -NH_3_^+^ form, which contradicts the HA_EDC_-DA/HA_Ox_-DA covalent bonding schemes proposed in the literature. The ^13^C ss-NMR results add useful new insights into the structure and interaction patterns of the conjugates. All of our findings are relevant for future practical applications, for instance in developing novel HA-based hydrogels. In addition, the present study demonstrates the importance of using the most appropriate analytical tools when investigating composite systems due to the complexity of hyaluronic acid conjugates. Solid-state NMR proved essential to answering the question in the title: actually, there is nothing wrong with hyaluronic acid chemistry; the claimed covalent bonds between DA and the HA(HA_Ox_) chain do not exist in these systems, because the conditions for their formation do not hold in practice.

## 1. Introduction

Owing to their outstanding hydrophilicity, biocompatibility, and mechanical properties, which are analogous to other biological soft tissues, hydrogels have attracted much attention in the biomedical engineering field [1,2]. Hyaluronic acid (HA) is among the most studied hydrogel-generating natural polymers, and consists of repeating disaccharide units of D-glucuronic acid and N-acetyl-D-glucosamine respectively coupled via alternating β-1,4 and β-1,3-glycosidic bonds [3,4]. In mammals HA is found in the form of hyaluronate, which occurs in high concentrations in different fibrous tissues, vitreous humor, skin, and synovial fluids [3] exhibiting physiological roles such as control of skin aging, inflammatory response, cartilage repair, lubrication of the corneal epithelium, and enhancing intestinal epithelial defense, to name only a few [5,6,7].

The linear polymeric molecule of HA is negatively charged at physiological pH, as the pKa of the HA carboxyl group is between 3 and 4 [3]. Due to numerous hydroxyl groups from the sugar moieties it is highly interactive with water molecules, decreasing its fluidity and influencing the movement and distribution of water [3,8]. Moreover, HA molecules are organized in different secondary structures depending upon concentration, pH, and molar mass, starting from rods or left single or double helices up to overlapped random coils [9] that are able to interact with numerous cell surface receptors. Thus, HA behaves as a signaling molecule that is able to regulate the adhesion, proliferation, and migration of the cells in different types of tissues [10]. In order to improve the mechanical properties of HA and augment it with new and unique properties, hybrids and conjugates of HA with different molecular systems have been recently developed, finding widespread applications such as bone tissue engineering, wound dressing, drug delivery, and infertility rehabilitation [11].

Motivated by the strong and almost substrate-independent adhesion of polydopamine (PDA), a material discovered in 2007 [12] by oxidative self-polymerization of dopamine (DA), the conjugation of HA with DA has attracted interest [13,14,15]. Upon subsequent oxidation, these conjugates have been shown to give rise to HA-based hydrogels with improved adhesion and mechanical stability. Even though the structure [16], polymerization, and adhesion mechanisms have not yet been fully elucidated, the range of PDA applications has greatly expanded in recent years [17] due to other favorable properties such as simple synthesis/deposition process, biocompatibility, and enhanced chemical reactivity of its catechol, amine, and imine groups. In this context, exploiting dopamine/polydopamine chemistry to develop better HA-based biomaterials is fully justified; the first step is to understand how dopamine interacts with HA under different synthesis conditions.

The most important HA-DA conjugates referred to in the literature are the HA_Ox_-DA conjugate, presented as a covalent HA-DA conjugate via Schiff base reaction between DA and dialdehyde-oxidized HA by sodium periodate [13], or the HA_EDC_-DA conjugate, which involves the covalent attachment of DA to HA via an amidic bond to the EDC/NHS activated carboxyl group of the D-glucuronic acid unit [14]. In both cases, covalent attachment of DA to the polysaccharide HA chain is presumed, while the conjugate is further stabilized via crosslinking of the catechol groups between chains. It is worth mentioning that such conjugate-based hydrogels exhibit superior adherence and improved mechanical resistance compared to native HA. Nevertheless, it is straightforward to ascertain that these types of conjugates have not been completely characterized; for instance, the covalent attachment of the DA to HA chain in the case of HA_EDC_-DA is judged either by ^1^H NMR spectroscopy in solution or by means of UV-vis and Fourier transform infrared (FT-IR) spectroscopy in the case of HA_Ox_-DA. These techniques do prove the presence of DA in the final hydrogel; however, they do not exclude the existence of a non-covalent physical attachment between HA and DA. On the other hand, solid-phase characterization of the final conjugation products, which could provide the exact covalent or non-covalent nature of the coupling between the components, has not been reported to date.

In the present study, we show that solid-state NMR spectroscopy can provide unique insights into the chemical structure of hyaluronic acid conjugates with dopamine, as it can directly indicate the presence or lack of a particular chemical bond or the charged/neutral form of a moiety while providing chemical site selectivity. We began our study by preparing a new conjugate by simply mixing sodium hyaluronate with dopamine hydrochloride at pH 5, which we refer to as HA-DA. Then, we performed a comparative ^15^N/^13^C ss-NMR analysis relative to the HA_EDC_-DA and HA_Ox_-DA conjugates. The results proved very insightful with respect to the way in which the components of the conjugates interact with each other depending on the particular synthesis conditions, at the same time revealing unexpected conclusions that contradict what has been reported previously in the literature on HA_EDC_-DA and HA_Ox_-DA.

## 2. Experimental

### 2.1. Materials

Sodium hyaluronate (MW: 1600–2500 kDa, injection grade, FS-HA-EP3.0, intrinsic 98 viscosity 2.7–3.6 m^3^/kg) was purchased from FreShine Chemicals (Las Vegas, NV, USA), dopamine hydrochloride (>99% purity), ethylene glycol (>99.5% purity), sodium periodate (extra pure grade), N-Hydroxysuccinimide (NHS, >99% purity), and dialysis tubing cellulose membrane (MWCO of 14 kDa) were purchased from Sigma Aldrich (St. Louis, MO, USA), and 1-(3-dimethylaminopropyl)-3-ethylcarbodiimide hydrochloride (EDC·HCl, >98% purity) was purchased from Thermo Scientific (Waltham, MA, USA). All chemicals were used as received without further purification.

### 2.2. Preparation of Hyaluronic Acid–Dopamine Conjugate (HA-DA)

The HA-DA conjugate was obtained by simple mixing of 500 mg sodium hyaluronate with equimolar dopamine hydrochloride in 50 mL phosphate buffered saline (PBS) at pH 5.0 for 16 h followed by dialysis against 5 L water three times over three days at room temperature. The clear dialyzed solution was frozen at −20 °C and lyophilized (Alpha 1-2 LDplus freeze-dryer from Martin Christ, Germany).

### 2.3. Preparation of NaIO_4_-Oxidized Hyaluronic Acid–Dopamine Conjugate (HA_Ox_-DA)

Oxidized hyaluronic acid (HA_Ox_) was obtained by dissolving 200 mg (0.5 mmoles) of sodium hyaluronate in 50 mL PBS pH 5.0, followed by the addition of solid sodium periodate in different molar ratio (1/1, ½, 1/3 and ¼) under vigorous stirring maintained for 48 hours at room temperature. The reaction was stopped by the addition of ethylene glycol in different volumes depending upon periodate content (0.75 mL for 1/1 reaction up to 3 mL for ¼ reaction). The obtained clear solution was dialyzed in cellulose tubing against 5 L water three times over three days at room temperature. The dialyzed solutions were frozen at −20 °C and lyophilized. The HA_Ox_-DA conjugate was then obtained by mixing 130 mg of 1/2 HA_Ox_ with 200 mg dopamine hydrochloride (DA·HCl) in 50 mL PBS pH 5.0 for 7 hours, yielding a whitish precipitate. The cloudy suspension was centrifuged for 20 minutes at 4000 RPM and the pellet was washed three times with 50 mL water with intermittent centrifugation, as mentioned above. The white precipitate was then lyophilized.

### 2.4. Preparation of EDC-Mediated Hyaluronic Acid–Dopamine Conjugate (HA_EDC_-DA)

To obtain the HA_EDC_-DA conjugate, 150 mg (0.375 mmoles) of sodium hyaluronate were dissolved overnight in 20 mL 2 × PBS pH = 5. 72 mg (0.375 mmoles) of EDC·HCl dissolved in 150 µL 2 × PBS, then 43.2 mg (0.375 mmoles) NHS dissolved in 100 µL 2 × PBS were added dropwise to the HA solution. After 20 minutes of stirring at room temperature, 75 mg (0.4 mmoles) of dopamine hydrochloride dissolved in 100 µL 2 × PBS was added to the reaction mixture. The pH value of the resulting solution was maintained between 4 and 6 for 9 h, followed by dialysis against 5 L acidified double-distilled water (DDW), 10 mM HCl, for 24 hours, which was repeated twice. The dialyzed solution was frozen at −20 °C and lyophilized to yield a white powder. This closely follows the procedure reported in [15], with the exception that we used the more readily available sodium hyaluronate. The two methods are almost equivalent judging from the relative concentrations of sodium and hyaluronic acid in the two cases.

### 2.5. Solid-State NMR Experiments

Solid-state NMR experiments were performed at room temperature on a Bruker Avance III spectrometer operating at 499.99 MHz ^1^H Larmor frequency. The ^13^C/^15^N ss-NMR spectra were acquired at a frequency of 14/7 kHz MAS (Magic Angle Spinning) using the Cross-Polarization (CP) pulse sequence. The CP contact times were set to 2 ms for ^13^C and 5 ms for ^15^N CP-MAS experiments. The spectra were recorded under high-power proton decoupling (100 kHz) by TPPM by averaging 80k (^13^C) and 140k (^15^N) transients, using a recycle delay of 2 s. The recorded ^13^C/^15^N NMR spectra were calibrated with respect to the CH_3_ line in TMS (tetramethylsilane) and the ^15^NO_2_ line in nitromethane through an indirect procedure which used α-glycine, Sigma Aldrich, 99% purity (176.5 ppm for the ^13^COOH line and −347.6 ppm for the ^15^NH_3_ line) as an external reference.

## 3. Results and Discussion

To verify the covalent bonding of DA molecule to the HA polymer chain claimed in the literature [13,15] for the HA_EDC_-DA and HA_Ox_-DA conjugates, we started by analyzing their ^15^N ss-NMR spectra. The proposed coupling mechanisms involve the DA amine site; thus, we expected that the ^15^N spectra would retrieve this information with the highest sensitivity. For this purpose, the spectra of the two compounds were compared with the spectra recorded on the reference HA, DA·HCl, and the HA-DA conjugate. As can be seen in Figure 1, the ss-NMR line of the amide nitrogen in the N-acetyl-glucosamine monomeric unit of HA is found at about the same spectral position in all the conjugates as in the pristine HA, −257.5 ppm: −258.5 ppm in HA-DA, −259 ppm in HA_EDC_-DA, and −260.3 in HA_Ox_-DA. This is not unexpected, essentially showing the marginal influence of DA conjugation on the HA amide regardless of the native or oxidized state of the polysaccharide chain. Note the asymmetric HA ss-NMR lineshapes in all the spectra, which are characteristic of amorphous polymeric systems.

The DA ^15^N ss-NMR line obtained on the reference DA·HCl at −336 ppm is characteristic of an -NH_3_^+^ amine group with a Cl^−^ ion in the close neighborhood. The DA NMR signal in the spectrum of the HA-DA conjugate comes from an -NH_3_^+^ moiety as well, now shifted to −347 ppm, which is very close to that found on glycine. Crystal packing in glycine is mainly driven by electrostatic interaction between the -NH_3_^+^ and -COO^−^ charged groups of adjacent molecules, suggesting that the DA^+^ molecules are most probably bound to the HA^−^ polymer chain in the HA-DA conjugate through non-covalent electrostatic interaction with the negatively charged carboxylate in the HA glucoronic acid monomeric unit.

Similarly, we found the DA ^15^N ss-NMR line in the HA_Ox_-DA conjugate at −341 ppm, again indicative of a free -NH_3_^+^ amine and not an imine moiety, for which typical chemical shift values should fall in the −30 to −100 ppm spectral range. Imine is predicted in [13] to form by covalent bonding of DA amine nitrogen to an aldehyde group on the HA_Ox_ chain through the Schiff base reaction. On the one hand, our results clearly contradict imine formation by covalent bonding of the DA molecule. On the other hand, our results show that if conjugation in HA_Ox_-DA is instead achieved via non-covalent interactions, there should be differences compared to the HA-DA case, as the measured DA ^15^N chemical shift here is between the values found in the DA·HCl and HA-DA cases.

The situation of the HA_EDC_-DA conjugate is more complex; here we obtain not one but three DA ^15^N ss-NMR lines. The two overlapped peaks at −342 ppm and −348 ppm correspond to charged -NH_3_^+^ amine in two distinct electronic environments, the former closer to the HA_Ox_-DA and the latter to the HA-DA conjugate. The occurrence of a third broader ^15^N peak centered on −285 ppm (marked with arrow in Figure 1) does in principle agree with the conclusion in [14,18] that covalent conjugation proceeds through the formation of an amide bond between DA molecules and the carboxylate groups in the HA glucoronic acid monomers (it approaches the range of typical chemical shift values for an amide nitrogen, −240 to −280 ppm [19]). Our results show, however, that even if this covalent conjugation mechanism works in HA_EDC_-DA, it involves only a fraction of the DA molecules. The rest are bound to the HA chain by means of non-covalent interactions, apparently close to those found in HA-DA and HA_Ox_-DA conjugates, which indicates the DA ^15^N chemical shift values of the -NH_3_^+^ amine obtained in all the three cases. 

To summarize, the results of the ^15^N ss-NMR analysis above appear to partially confirm, i.e., only for the HA_EDC_-DA conjugate, the covalent bonding of dopamine to hyaluronic acid through its primary amine of nitrogen, whereas the ubiquitous presence in all the three investigated conjugates of resonance peaks in the −341 to −348 ppm range, assigned to -NH_3_^+^ nitrogen, points rather towards non-covalent electrostatic interactions between this charged moiety and the HA chain. However, the observed variation of almost 7 ppm in the -NH_3_^+^ chemical shifts suggests that the DA interactions with the polymeric backbone are not identical in all the three conjugates; thus, a ^13^C ss-NMR analysis was performed in the hope of obtaining complementary information able to clarify this issue.

The conjugates obtained from unmodified HA, that is, HA_EDC_-DA and HA-DA, are analyzed first in Figure 2 by comparing their ^13^C ss-NMR spectra with the spectra recorded on DA∙HCl and HA. The assignment of the ^13^C ss-NMR lines to the various carbon positions in these two reference compounds is shown directly in the figure. All eight distinct dopamine carbon sites in DA∙HCl provide well-resolved lines in the ^13^C ss-NMR spectrum, and as such are explicitly assigned by means of the corresponding labels in the displayed chemical structure. For HA, we used the following convention: (i) the labels corresponding to the different carbon sites in the glucoronic acid and N-Acetyl-Glucosamine monomeric units of HA are represented in red and blue, respectively; (ii) only the sidechain carbons are labeled explicitly in the drawn chemical structure, whereas for the C1–C5 carbons of the glucose rings the conventional labeling scheme is assumed; and (iii) the most intense peak at 76 ppm in the HA ^13^C ss-NMR spectrum is a superposition of the lines from all the carbon sites that have not been explicitly assigned to the other smaller peaks.

The analysis of the information contained in the ^13^C ss-NMR spectra of the HA-DA and HA_EDC_-DA conjugates is greatly facilitated by the fact that there is no spectral overlapping of the NMR lines generated by the two component systems (i.e., HA and dopamine). From this perspective, it is obvious that HA-DA is a simple physical association of the two subsystems, which are linked together by non-covalent interactions. The HA peaks in HA-DA are essentially unaltered compared to the peaks in the HA spectrum, whereas the dopamine lines are only broadened compared to their counterparts in DA∙HCl while maintaining almost the same spectral positions. The only exception is the DA C3 carbon line, for which a small 2 ppm shift is observed, most probably caused by a change in the relative conformation of the ethylamine sidechain with respect to the catechol ring. As explained above, this broadening is a consequence of changing the intermolecular environment for the DA^+^ molecules from an ordered crystalline network to a disordered distribution in an amorphous composite system.

Moving next to the case of HA_EDC_-DA, it can be seen that the ^13^C ss-NMR spectrum contains more resonance lines than would be expected from a simple superposition of the HA and DA spectra. The aliphatic C1 and C2 and the aromatic C3, C4, C7, and C8 dopamine NMR lines are found at about the same positions as in HA-DA, except that they are broader and have lower intensities. The C5 and C6 -C-OH signal, which in HA-DA spectrum provides the intense line at 145 ppm, is instead either of a very low intensity or split into two lines, as in DA∙HCl. The second assumption is, however, less probable, considering the larger separation of about 10 ppm between these two lines in HA_EDC_-DA compared to only 3.5 ppm in DA∙HCl. Thus, most likely one of the ^13^C ss-NMR peaks at 139 and 148.8 ppm is produced by another chemical species present in HA_EDC_-DA, a conclusion which is supported by the additional lines found at 14.8 ppm, 38.9 ppm, and 159.2 ppm (marked with arrows in the figure) that are certainly not produced by dopamine. In fact, these extra lines are rather compatible with the formation of acylurea adducts of EDC with HA, as demonstrated in [20], because they can all be assigned to chemical groups present in such adducts (methyl, methylene, -C=O, and -C-O-). This prompts us to re-assign the ^15^N ss-NMR spectrum of HA_EDC_-DA in Figure 1; taking into account the ^13^C ss-NMR results, the broad peak spanning the spectral window from −295 to −280 ppm is in fact better explained by a superposition of lines determined by the multiple tertiary amines in EDC than by a single DA amide nitrogen line, further supporting the conclusion that EDC remains incorporated into the HA_EDC_-DA conjugate. Both results strengthen one of the findings in [20], namely, that “the reaction of the carboxyl of glucoronate residues with EDC in the presence of primary amines yield only acylurea adducts rather than the expected amide coupling products”.

The last case to discuss is the HA_Ox_-DA conjugate, again by comparing its ^13^C ss-NMR spectrum with the reference spectra of DA∙HCl and HA_Ox_ (see Figure 3). Before proceeding to the analysis, it is important to first draw attention to a misconception that tends to be perpetuated in the recent literature regarding the structure of NaIO_4_ oxidized hyaluronic acid [14,18,21,22,23,24]. It has been claimed in several works that HA oxidation leads to ring opening of the glucoronic acid monomer and the formation of dialdehydes at the bond cleavage sites. Others, especially in earlier works, have demonstrated that the aldehyde groups are unstable and transform to hemiacetals and/or aldols [25,26]. This is normally evidenced using ^13^C ss-NMR by the missing of aldehyde peaks at chemical shifts larger than 190 ppm, which is obviously the case for the HA_Ox_ spectrum in Figure 3. The distinctive signature of the aldehyde transformation to hemiacetals is the occurrence of new ss-NMR peaks in the range of 91–93 ppm. This feature, as well as an analysis of other small changes observed in the HA_Ox_ ^13^C ss-NMR spectrum relative to HA as a function of the oxidation degree, including a comparison with simulated spectra, are all presented in the Appendix A.

Having clarified these details about the HA_Ox_ spectrum, we can proceed to discussing the HA_Ox_-DA conjugate. A close inspection of the experimental data reveals both similarities and differences compared to the case of HA-DA, although their ^15^N ss-NMR spectra indicate a certain resemblance between the two systems. First, it should be noted that the DA ss-NMR peaks in the HA_Ox_-DA spectrum show both a broadening with respect to the DA∙HCl lines and other changes, the most prominent being the C3 line-shift from 132 ppm to 125.5 ppm, larger than in the HA-DA case, and a substantial reduction in the intensity of the C2 signal. The latter could in principle be correlated with an intensity increase in the 58–59 ppm range of the HA_Ox_-DA spectrum if we assume that some of the DA molecules in this composite have the C2 carbon ss-NMR line shifted to this new spectral position (marked with arrow in Figure 3).

Our ss-NMR experimental findings on HA_Ox_-DA allow us to clarify a few structural features; unfortunately, they cannot provide complete structural characterization that was accomplished for HA-DA. On the one hand, the covalent bonding of the DA through its nitrogen can be excluded based on the ^15^N ss-NMR spectrum and the fact that the C1 ^13^C ss-NMR line has almost the same position in both the DA∙HCl and HA_Ox_-DA spectra. On the other hand, the analysis of the HA_Ox_-DA ^13^C ss-NMR spectrum clearly points to the conclusion that interaction of DA with the HA_Ox_ chain markedly changes the electronic environment around the DA C2 and C3 carbon positions. However, whether this change is due to covalent bonding through the C2 carbon, for instance via -C-O-C- bridges with hemiacetals, or is a simple non-covalent physical interaction, cannot be further speculated on here, as we do not have additional information to strengthen either of these two hypotheses. 

## 4. Conclusions

A thorough ss-NMR investigation was performed on three distinct dopamine conjugates of hyaluronic acid, both in its native and NaIO_4_-oxidized form. We unexpectedly found that in all these conjugates the DA molecules are always in their charged -NH_3_^+^ form, as clearly evidenced from the ^15^N ss-NMR spectra. This suggests non-covalent bonding of DA to the polysaccharide chain, with the electrostatic interaction playing a major role. The most straightforward case is that of HA-DA, a conjugate reported for the first time in this work, obtained by mixing dopamine hydrochloride with sodium hyaluronate at pH 5. Here, the ^15^N and ^13^C ss-NMR spectra are compatible with a composite system in which the charged DA^+^ molecules simply replace the Na^+^ ions near the -COO^−^ groups of the HA glucoronic acid monomeric units. For the other two conjugates, the combined ^13^C/^15^N ss-NMR analysis did not reveal such a complete interaction pattern, although it did provide useful insights into the structure of the formed assemblies. The HA_EDC_-DA conjugate [14] turned out to be a tri-component system, with the EDC incorporated into HA as acylurea adducts and the DA^+^ molecules interacting non-covalently with them. The HA_Ox_-DA conjugate [13] is the most puzzling; whereas the ^15^N ss-NMR spectrum indicates a free DA molecule with its -NH_3_^+^ line very close to that found in DA∙HCl, its ^13^C ss-NMR spectrum shows marked changes of the DA C2 and C3 lines, indicating that these molecular sites are affected by the interaction with the HA_Ox_ chain. This interaction most probably involves hemiacetals formed along the oxidized HA chain, as the existence of such groups represents the only difference between the HA-DA and HA_Ox_-DA conjugates. One possibility would be the covalent bonding of DA to hemiacetals via -C-O-C- bridges; however, the available ss-NMR data are not sufficient to fully confirm this hypothesis, and further work is needed to clarify the issue. 

Having established all these details, we are now at the point of answering the question in the title. The key result of the present work is that regardless of the exact nature of the interaction between the DA molecules and the HA/HA_Ox_ chains in the HA_EDC_-DA/HA_Ox_-DA conjugates, their claimed [13,14] covalent bonding through the amine group does not take place, even though such bonding is plausible from a chemical standpoint. Our ss-NMR study shows that there is nothing wrong with hyaluronic acid chemistry; this simply happens because the conditions for the formation of such covalent bonds are not met in practice. The recorded ss-NMR spectra actually confirm what was already known from the older literature: EDC cannot mediate amide bond formation between DA and HA because it remains covalently bound to HA [20], whereas the Schiff base reaction between DA and the aldehyde groups of HA_Ox_ is impossible because there are no stable aldehydes in HA_Ox_, these moieties being transformed into hemiacetals [25].

On practical side, the present study introduces the combined ^15^N/^13^C ss-NMR as a fast and reliable, method for identifying the nature of the coupling between primary amines and hyaluronic acid. This information can be extracted by direct visual inspection of the ^15^N ss-NMR spectrum through the shift of the original amine line(s). Furthermore, this study shows that in addition to the commonly employed solution-state ^1^H NMR in synthetic chemistry, a multi-nuclear investigation (^15^N/^13^C) in either solution or solid state can often be required to unambiguously constrain the nature of the formed bonds between the component sub-systems. Establishing the nature of the coupling is relevant for the subsequent use of such complexes for the formation of hydrogels, revealing whether they can form chemically or only physically cross-linked hydrogels.

## Figures and Tables

**Figure 1 polymers-15-02825-f001:**
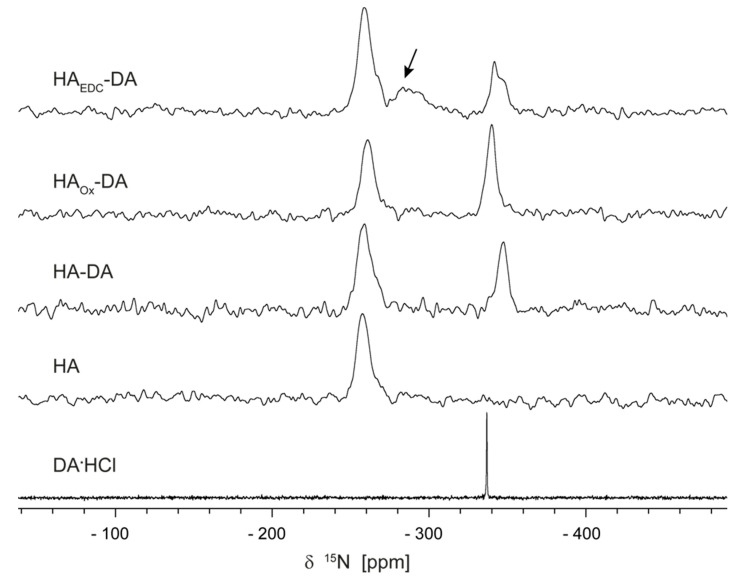
The ^15^N ss-NMR spectra of DA∙HCl, HA, HA-DA, HA_Ox_-DA, and HA_EDC_-DA recorded by the CP-MAS sequence as described in Section 2. Marked with arrow in the HA_EDC_-DA spectrum is a broad ss-NMR peak not found in the case of the other two HA composites.

**Figure 2 polymers-15-02825-f002:**
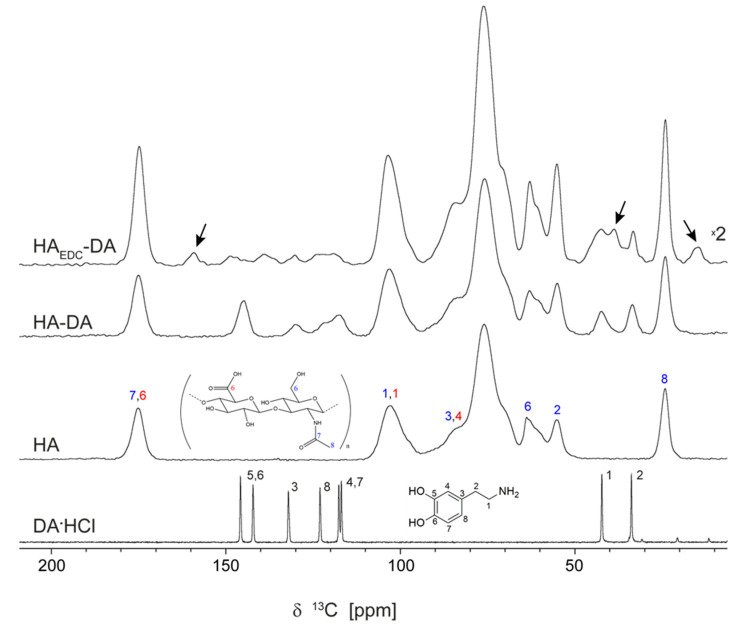
The ^13^C ss-NMR spectra of DA∙HCl, HA, HA-DA, and HA_EDC_-DA recorded by the CP-MAS sequence, as described in Section 2. The assignment follows the labeling scheme in the drawn chemical structures, which is detailed in the text. Those ss-NMR lines certainly not produced by DA are marked with arrows in the HA_EDC_-DA spectrum.

**Figure 3 polymers-15-02825-f003:**
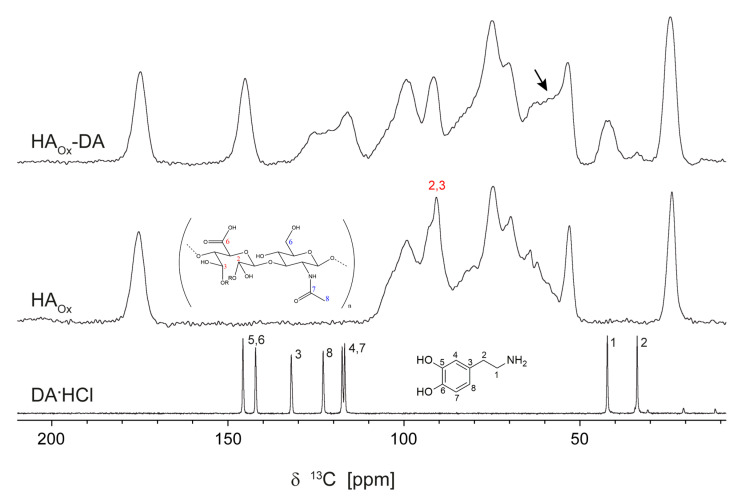
The ^13^C ss-NMR spectra of DA∙HCl, HA_Ox_, and HA_Ox_-DA recorded by the CP-MAS sequence as described in Section 2. The assignment follows the labeling scheme in the drawn chemical structures, which is detailed in the text; compared to pristine HA in Figure 2, only the new peaks corresponding to hemiacetal formation are explicitly shown here. The possible line-shift of an important part of the DA C2 carbons is marked with an arrow in the HA_Ox_-DA spectrum.

## Data Availability

Raw data, ss-NMR spectra, are available upon request from the corresponding author.

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
