# Peer review of "What Is Wrong with Hyaluronic Acid Chemistry? A 15N/13C Solid-State NMR Re-Evaluation of Its Dopamine Conjugates"

_polymers, 2023, doi:10.3390/polym15132825_

Round 1

Reviewer 1 Report

Comments:

1. The previous other techniques or typical NMP, FT-IR and mass spectroscopy should be addressed as literature review for identifying these three dopamine (DA) conjugates of hyaluronic acid. The limitation of them for applying should be discussed.

2. Schematic diagram presenting synthesis processed should be included in Method for easier understanding of reader.

3. Chemical structure of three dopamine (DA) conjugates of hyaluronic acid should be revealed in Introduction.

4. Please check Glycine in line 157 that should be glycine? and Glucoronic in line 161 that should be glucoronic?

5. More discussion with supporting refs should be included in paragraph 1-3, 4-6, and others for strengthen the discussion and clearly understanding.

6. The suggestion from your finding for the further investigation or application should be added in Conclusion.

7. Please consider for incorporate the schematic diagram of your novelty of finding in Conclusion.

There is no problem relied on the language; nevertheless, only some points should be corrected as mentioned in comment to authors.

Reviewer 2 Report

The manuscript entitled “What is wrong with Hyaluronic Acid chemistry? A solid-state NMR re-evaluation of its Dopamine conjugates” describes the preparation of an association complex between HA and DA, and two conjugates of the same molecules according to literature methods. The work is mainly devoted to the characterization of the outcomes of the different preparation routes by means of solid state 15N and 13C NMR, which reveal that the employed preparation methods for the conjugates, contrary literature claims, are far from successful.

The authors suggest that the the erroneous literature reports are due to not suited characterization methods that prevented the researchers to recognize the true nature of the supposed conjugates.

Most commonly the hyaluronic acid conjugates are characterized by FTIR spectra in the solid state and 1H NMR spectra in solution. Probably, a too hastily examination of the latter, relying just on the presence of signals of both the polysaccharide and the small molecules can lead to erroneous conclusions about the tre nature of intermolecular interactions. Thus, an extremely accurate examination of 1H and further information from 13C, and possibly, 15N NMR is required to a precide definition of the actually obtained products.

The authors employ 15N and 13C NMR in the solid state, but it most considered that solid state NMR facilities are not as easily accessible as solution ones.

An obvious question is : Is it possible to run the 13C and 15N NMR spectra, maybe 2D inverse ones, on the specimens subject of the present contribution?

I suggest the authors to run at least the 1H NMR spectra in solution, and to correlate them with the information provided by the solid state NMR. If it is possible to record 1H NMR spectra in solution, very helpful, in order to discriminate between covalent and not covalent interactions, are diffusion measurements through PGSE NMR experiments.

Furthermore, since ref 20 is particularly relevant to the discussion, I suggest to exploit it more extensively do describe better the present results.

I suggest to modify the title to make it more informative, adding 15N and 13C before “NMR” and association complexes after “conjugates”

Unfortunately, the examined products were mainly association complexes rather than true conjugates. To assess the effectiveness of solid state 15N NMR to distinguish between true covalent conjugates and non covalent association products, it might be interesting to examine a successful example of covalent amine derivatization, such as the HA−spermidine derivative, extensively characterized by 13C and 1H NMR in solution and also by solid state 13C CP-MAS NMR spectroscopy, after cross-linking, as reported in

Synthesis and NMR Characterization of New Hyaluronan-Based NO Donors C. Di Meo, D. Capitani, L. Mannina, E. Brancaleoni, D. Galesso, G. De Luca, and V. Crescenzi
Biomacromolecules 2006 7 (4), 1253-1260 DOI: 10.1021/bm050904i

Round 2

Reviewer 2 Report

In the revised version of the manuscript “What is wrong with Hyaluronic Acid chemistry? A 15N/13C solid-state NMR re-evaluation of its Dopamine conjugates” the examined nucleiare mentioned in the title and there are two new paragraphs.

I realize that we are always in a hurry and probably this is the reason why synthetic chemists rely on hastily interpretyed 1H solution NMR spectra, e.g., ref.13, and do not take advantage of multinuclear solid state NMR, in spite ref. 20 appeared in 1992.

Author Response

Thank to the reviewer for pointing out that aspect: we added another phrase in the conclusion to introduce that very useful suggestion.